# Metabolic Biomarkers of Liver Failure in Cell Models and Patient Sera: Toward Liver Damage Evaluation In Vitro

**DOI:** 10.3390/ijms252413739

**Published:** 2024-12-23

**Authors:** Simone Rentschler, Sandra Doss, Lars Kaiser, Helga Weinschrott, Matthias Kohl, Hans-Peter Deigner, Martin Sauer

**Affiliations:** 1Institute of Precision Medicine, Furtwangen University, Jakob-Kienzle-Straße 17, 78054 VS-Schwenningen, Germany; 2Fraunhofer Institute IZI (Leipzig), Department Rostock, Schillingallee 68, 18057 Rostock, Germany; 3Faculty of Science, Tuebingen University, Auf der Morgenstelle 8, 72076 Tübingen, Germany; 4Department of Anesthesiology and Intensive Care Medicine, University Hospital of Rostock, Schillingallee 35, 18057 Rostock, Germany; 5Center for Anesthesiology and Intensive Care Medicine, Hospital of Magdeburg, Birkenallee 34, 39130 Magdeburg, Germany

**Keywords:** in vitro cell models, drug toxicity testing, metabolic biomarker, hepatocellular injury, in vivo–in vitro translation

## Abstract

Recent research has concentrated on the development of suitable in vitro cell models for the early identification of hepatotoxicity during drug development in order to reduce the number of animal models and to obtain a better predictability for hepatotoxic reactions in humans. The aim of the presented study was to identify translational biomarkers for acute liver injury in human patients that can serve as biomarkers for hepatocellular injury in vivo and in vitro in simple cell models. Therefore, 188 different metabolites from patients with acute-on-chronic liver failure before and after liver transplantation were analyzed with mass spectrometry. The identified potential metabolic biomarker set, including acylcarnitines, phosphatidylcholines and sphingomyelins, was used to screen primary and permanent hepatocyte culture models for their ability to model hepatotoxic responses caused by different drugs with known and unknown hepatotoxic potential. The results obtained suggest that simple in vitro cell models have the capability to display metabolic responses in biomarkers for liver cell damage in course of the treatment with different drugs and therefore can serve as a basis for in vitro models for metabolic analysis in drug toxicity testing. The identified metabolites should further be evaluated for their potential to serve as a metabolic biomarker set indicating hepatocellular injury in vitro as well as in vivo.

## 1. Introduction

Liver disease remains a significant health issue with increasing importance anticipated in many countries [1,2]. Liver disease and cirrhosis account for 2 million deaths worldwide each year, which are caused by a variety of factors including genetic causes, autoimmune disorders, alcohol consumption, viral infections, and non-alcoholic fatty liver [3,4,5]. Complications of chronic liver disease and cirrhosis include hepatic decompensation, acute kidney injury and acute-on-chronic liver failure [3]. Acute-on-chronic liver failure combines acute hepatic decompensation and liver and extrahepatic organ failure and usually involves a provoking event such as surgery, sepsis, ischemia, or superimposed liver injury from alcohol, hepatotoxic drugs, or hepatitis virus infection [6,7].

Acute liver failure, on the other hand, refers to a rare but severe condition resulting from abrupt hepatocyte injury. Causes include toxins, hepatitis (with viral or autoimmune origin), hepatic ischemia and drug-induced liver injury (DILI) [8,9,10]. 

Over 50% of all acute liver failures are caused by drugs with acetaminophen accounting for the largest share [11]. Nevertheless, more than 1000 drugs, toxins, over-the-counter herbal medicines and dietary supplements are known for having a potential to induce liver injury [12,13,14,15]. An early identification of potential hepatotoxic drugs during development and preclinical testing is, therefore, essential to minimize health risks for the patients [16]. 

Traditionally, necessary safety tests to evaluate chemical toxicity are performed using animal models, especially rats and mice. However, in addition to concerns about animal welfare, these studies are costly and time consuming [17]. Moreover, animal models often fail to predict human hepatotoxicity due to species-specific differences in (drug) metabolism and toxicity targets [16]. In a retrospective study of 150 drugs, only 43% of human toxicities had been correctly predicted with animal tests compared to clinical experience [18]. Accordingly, cultures of primary human hepatocytes (PHHs) and human hepatoma cell lines are now routinely used in addition to in vitro models for studying drug metabolism and for toxicity testing [19,20,21]. In in vitro cell culture toxicity assays, screenings for the presence of specific metabolites of the tested drugs or generated metabolites by the drugs/drug interactions are performed. However, this screening approach for single-biomolecule-based endpoints can only be applied if there is knowledge on the possible mode of action of investigated drug [22]. Other tests applied in in vitro toxicological are cell viability assays that determine toxicity endpoints. Here, functional assays rely on the measurement of one or more cytotoxic indicators (e.g., loss of membrane integrity, apoptosis [23].

However, there is strong evidence that in vitro cell-based assays are yet not fully sufficient for displaying the whole hepatotoxic potential of tested chemicals, as substantial percentages of new drugs fail late-stage human drug testing, receive a regulatory “black box” warning for causing drug-induced liver injuries, or are removed from the market for safety reasons [16,24,25].

Given that the mechanisms involved in adverse reactions to xenobiotics and DILI are complex and often involve multiple biological processes and/or organ systems, more specific markers of toxicity and mechanism-based combinations thereof are necessary [22,26].

Metabolic biomarkers are well suited to serve as prognostic, predictive and diagnostic biomarkers for multiple conditions and have several advantages over gene transcripts or proteins as biomarkers [27,28,29,30,31].

Since the early 2010s, there has been a growing interest in in vitro investigations of hepatotoxicity using experimental metabolomics setups. Among these, the HepG2 cell line is the most popular cell model; however, primary hepatocytes from humans (PHHs), rodent and fish as well as a wide variety of other permanent cell lines have been employed also [26]. Nevertheless, PHHs are frequently preferred as in vitro models in toxicity and metabolism studies [32], since they better represent structural and biochemical processes found in vivo [25]. Their use, however, is limited by high costs, short lifespan, and poor availability as well as the significant variability between hepatocyte preparations of different donors [33,34,35]. On the other hand, human hepatoma cells provide unlimited lifespan, stable phenotype and easy handling [16,36]. These cell lines are useful for studying the liver function and general mechanisms of toxicity, whereas drug metabolism and toxicity predictions may not be adequately detected due to a lack of some metabolic enzyme families and, respectively, lower levels of enzymes than in PHH [37,38].

The identification of safety biomarkers that exhibit similar responses in different species, thus enabling a comparison of nonclinical studies with clinical studies, is of great importance for studies on drug development. Ideally, the translational safety biomarkers provide more sensitive and specific information than current clinical chemistry biomarkers. They can be assessed in all species commonly employed in safety assessment (mouse, rat, dog, non-human primate), can be measured noninvasively or in accessible fluids (blood, urine), are specific for organ injury or toxicity mechanisms, and are insensitive to non-toxic perturbations (age, diet, other disease) [39,40,41]. Clinical safety biomarkers have been identified and qualified for use in nephro- and cardiotoxicity. There are, however, ongoing projects for the identification of safety biomarkers for liver, vascular injury and skeletal muscle [40].

Approaches for the reverse translation of biomarkers from patients to nonclinical applications have already been employed for circulating biomarkers for drug-induced cardiotoxicity [23]. Most of these approaches concentrate on biomarkers that are already used in clinical settings given the great advantage of existing clinical acceptance for the diagnosis and prognosis for these biomarkers [42].

However, as changes in potential biomarkers (e.g., RNA/protein) often differ in animals and existing cell cultures from changes in humans, the transferability of the results of in vitro assays and identified biomarkers remains difficult in many cases. To overcome the obstacle of interspecies differences in translational biomarker identification, this study aims to first identify potential biomarkers in human patient samples of hepatocellular injury and then verify their ability to indicate hepatocellular injury in in vitro cell culture systems during drug toxicity screenings. Therefore, 188 different metabolites are analyzed in plasma samples from patients with acute-on-chronic liver failure before and after liver transplantation using targeted mass spectrometry, and significantly altered metabolites are identified. The biomarkers for hepatocellular injury identified using this approach are then transferred to simple in vitro cell models. Here, the biomarkers are evaluated for their ability to make meaningful predictions about drug-induced hepatocellular injury in simple cell models of primary and permanent hepatocyte cultures.

## 2. Results

### 2.1. Survival, Laboratory Parameters, Cytokines and Clinical Characteristics of Patients

In the clinical study, 11 patients suffering from acute-on-chronic liver failure before and after liver transplantation were included (liver-transplantation group). As the control cohorts, there were 21 postoperative patients (postoperative-control group) without known liver disease and 14 healthy control subjects. Summaries of laboratory parameters, clinical characteristics and the results of the APACHE II, SOFA, Child–Pugh scores at inclusion are displayed in Table 1.

Only one patient of the liver-transplantation group died (in-hospital mortality), and all patients of the postoperative-control group survived. In the postoperative-control group, the average of age was higher than in the liver-transplantation group, and predominantly female patients were included in the study (both groups). In the liver-transplantation group, all patients improved dramatically in their clinical status after transplantation, because all organs started to work immediately after surgical treatment.

All values of TNF-alpha, IL-10 and IL-6 were low in the liver transplant group before surgical treatment (Table 2). Two hours after liver transplantation, a significant increase in IL-6 and IL-10 was observed; the cytokine levels normalized after 12 and 24 h but rose slightly again after 168 h of surgical treatment. TNF-alpha levels were always low, and a slight increase was only observed 2 h after transplantation.

### 2.2. Identification of a Potential Metabolic Biomarker Set for Liver Cell Damage

Samples from patients with acute-on-chronic liver failure before and at different time points after liver transplantation were investigated for specific metabolite changes to identify a potential metabolic biomarker set for hepatocellular injury. After removing plate effects, 141 metabolites were considered for further analysis. Log fold changes in metabolite concentrations in LT patients, post-OP control patients and healthy control patients are displayed as a heatmap in Figure 1A. We identified 35 metabolites being significantly altered in patients with hepatocellular injury compared to healthy control patients but not in post-OP control patients compared to healthy controls (Figure 1B). Among these, free carnitine as well as multiple short-, middle- and long-chain acylcarnitines were increased, whereas several phosphatidylcholines and sphingomyelines were found to be decreased.

The metabolic analysis of patients revealed significant elevated levels of several short-, medium- and long-chain acylcarnitines (C0, C2, C3, C3.OH, C4, C3.DC…C4.OH., C4.1, C5, C5.OH…C3.DC.M., C5.1, C5.DC…C6.OH., C6.1, C7.DC, C8, C10, C10.1, C14, C14.1.OH, C14.2.OH, C16, C16.OH, C16.1, C16.1.OH, C16.2, C16.2.OH, C18.1, C18.1.OH, C18.2) as well as phosphatidylcholine PC.aa.C32.0 and decreased levels of several other phosphatidylcholines (PC.aa.C34.1, PC.aa.C38.6, PC.aa.C40.5, PC.aa.C40.6) and sphingomyelines (SM.C16.1, SM.C20.2). The log2-fold changes of metabolites in patients with liver failure at the time points studied are presented in Figure 2. Interestingly, the concentrations of some metabolites were found to return to levels comparable to those of healthy control patients.

### 2.3. Metabolic Changes in Response to Drug Treatment in Primary and Permanent Hepatocytes

The hepatocellular injury-specific metabolites identified in vivo in humans were assessed for their ability to make meaningful predictions about drug-induced hepatocellular injury in simple cell models. Therefore, primary human hepatocytes and HepG2/C3A cells were exposed for two times for 72 h to different concentrations of several drugs. Afterwards, the 188 metabolites previously analyzed in the patients were then analyzed in the cells.

Importantly, a large proportion of the measured metabolites (147 out of 188) was strongly biased by the corresponding donor, leading to the exclusion of a significant part (21 out of 35) of the previously identified metabolite set during the analysis. In response to hepatotoxic APAP exposure, seven metabolites of the biomarker set were significantly altered in PHH cells. Six of the significant metabolites (C2, C6.1, C16.OH, C16.2, C16.2.OH, PC.aa.C38.6) showed a similar behavior to that observed in patients. In contrast, the metabolite PC.aa.C32.0 was significantly altered in the opposite direction to that observed in patients. Log fold changes in metabolite concentrations in PHH are displayed as a heatmap in Figure 3A. The specific metabolites were also altered in response to the exposure with the drug rocuronium (ROC) in C_max_ and 10× C_max_ concentrations. However, only a small proportion of these metabolites were found to be significantly altered after exposure to different concentrations of sufentanil (SUF) or fluconazole (FLUCO, Figure 3B).

In contrast to PHH cells, no metabolites had to be excluded during the analysis of HepG2/C3A cells, enabling the consideration of the complete metabolite set from patient data. Log fold changes in metabolite concentrations in HepG2/C3A are displayed as a heatmap in Figure 4A. Following APAP exposure in hepatotoxic concentrations, 15 of the 35 metabolites demonstrated significant changes, while C2, C3, C3.OH, C3.DC…C4.OH, C4.1, C5, C5.DC…C6.OH, C8, C10, PC.aa.C34.1, PC.aa.C38.6, PC.aa.C40.5, PC.aa.C40.6 and SM.C16.1 demonstrated a similar response as in patients, and the metabolite PC.aa.C32.0 again demonstrated a opposite response (Figure 4B). This effect could also be observed in response to the treatment with different concentrations of rocuronium. Following the exposure with different concentrations of sufentanil or fluconazole, however, only a low amount of these metabolites were found to be significantly changed. Furthermore, several of the significant metabolites demonstrated an opposite response compared to the patient data set.

More metabolites were significantly altered in response to the exposure to different concentrations of anidulafungin (ANIDU) and caspofungin (CASPO). Most of the acylcarnitine alterations were comparable to those in patients, whereas phosphatidylcholines and sphingomyelines demonstrated a different behavior. Interestingly, log2 fold changes in the metabolite concentrations are similar for the exposure with C_max_ and 10× C_max_.

An overview of the metabolites with significant log2 fold changes in the patients before liver transplantation and in the two cell culture models in response to the exposure with several drugs at different concentration is provided in Table 3.

## 3. Discussion

Liver failure remains a serious complication with a high mortality rate and comparatively high incidence (approx. 11% of all intensive care patients and around 19% in septic shock). It can occur acutely or as an exacerbation of chronic liver disease. The etiology is multifactorial and includes pre-existing liver disease as well as acute clinical events such as infection, hemorrhage and the use of hepatotoxic drugs. Drug-induced liver injury (DILI) is particularly common, with paracetamol being the most frequent trigger, which is followed by antibiotics and antiepileptic drugs. The pathophysiology of DILI includes direct hepatotoxic effects and immune-mediated reactions with a spectrum ranging from asymptomatic liver enzyme elevations to fulminant liver failure [43,44,45]. The early identification of potentially hepatotoxic drugs during development and preclinical testing is, therefore, crucial for minimizing health risks to patients. The objective of the presented study was to identify translational biomarkers of acute liver injury in humans, which could be used to predict hepatocellular injury in vivo and in vitro in cell models during drug development.

In this study, we investigated changes in metabolites in patients suffering from acute-on-chronic liver failure with different pathogenesis and evaluated the metabolic changes in primary and permanent human hepatocyte cultures in response to the treatment with different drugs with known (acetaminophen) and unknown hepatotoxic potential (rocuronium, sufentanil) as well as drugs for which rare cases of hepatotoxic events had been reported (fluconazole, anidulafungin, caspofungin). We here describe two major findings. First, a potential metabolic biomarker set for hepatocellular injury was identified in plasma samples from patients with an acute-on-chronic liver failure before and after liver transplantation. Importantly, concentrations of most of the affected metabolites returned to levels in healthy controls within several hours after liver transplantation, implying a relation to liver function. Second, several of these metabolites were also found to be significantly changed in the in vitro cell models treated with the hepatotoxic compound acetaminophen. These changes were detected in both PHH and HepG2/C3A hepatocytes with HepG2/C3A better representing metabolic responses.

These metabolites, therefore, represent potential biomarkers for the prediction, diagnosis and prognosis of diseases with hepatocellular injuries and may also serve as translational biomarkers for in vitro assays of drug toxicity. The corresponding set for hepatocellular injury in vivo and in vitro contains free carnitine, different acylcarnitines, phosphatidylcholines and a sphingomyeline.

The identified biomarker set can reflect different cellular mechanisms in humans due to its diversity. These include mechanisms that are currently considered to play a decisive role in drug-induced liver insufficiency (DILI) [46,47,48].

Direct hepatotoxicity in DILI is frequently caused by the interaction of the drug or its metabolites with cellular macromolecules leading to protein dysfunction, DNA damage, lipid peroxidation and oxidative stress [11]. The peroxidation of the unsaturated fatty acids of the endoplasmic reticulum leads to fatty degeneration of the liver, damaging the mitochondria and biomembranes. Fatty degeneration of the liver therefore also leads to an increased formation of mitochondrial oxygen radicals, which leads to the inhibition of β-oxidation. This in turn can lead to steatohepatitis and fibrosis or cirrhosis [49].

Moreover, the mitochondrial function and energy production may also be disrupted by defective ionic gradients and intracellular calcium stores [11]. As a result of oxidative stress, glutathione is depleted in the hepatocytes, reducing antioxidant capacity and leading to further cell damage [49]. Fatty degeneration and cell necrosis may also be caused by the inhibition of the activity of RNA polymerases II and III. The inadequate function of polymerases generally affects the synthesis of enzymes, structural proteins and apolipoproteins [49].

Therefore, to serve as safety biomarkers for DILI in the assessment of drug toxicity, the biomarkers should ideally cover the major processes: mitochondrial dysfunction, inhibition of β-oxidation, oxidative stress and (lipo)apoptosis.

Free carnitine (C0) and the acylated derivatives (ACs) are essential for normal mitochondrial function. Fatty acids are transported across mitochondrial membranes via the carnitine shuttle system for fatty acid β-oxidation (FAO). During this process, acylcarnitine derivatives are generated by the transacylation of acyl-CoA species to free carnitine at the outer mitochondrial membrane [50,51]. For normal biological functionality, it is important that the concentrations of carnitine and acylcarnitines are sustained within relatively narrow ranges and corresponding alterations indicate impaired FAO and mitochondrial dysfunction [52,53]. In line, alterations in blood carnitine levels have been described for several medical conditions as end-stage renal failure and chronic liver diseases including liver cirrhosis [54,55,56,57]. Moreover, carnitine levels were found to be gradually more elevated with increasing degrees of hepatic affection [55]. Our results also demonstrated an increase in the plasma levels of free carnitine and a broad spectrum of acylcarnitines. These findings are consistent with a study using mice with induced acute liver failure investigating carnitine and acylcarnitine profiles. Here, mitochondrial dysfunction was observed with increased serum levels of short-, medium- and long-chain acylcarnitines [58]. Increased free carnitine levels were also found in a bosentan-induced liver toxicity model using human hepatoma HepaRG cell cultures [59].

Acylcarnitines, particularly medium-chain fatty acid acylcarnitines (MCACs), have also been shown to play a role in promoting inflammation by activating key pro-inflammatory pathways [60,61]. Studies using murine RAW 264.7 macrophages indicate that C12- and C14-carnitines activate NF-κB, displaying an critical inflammatory signaling node [62]. Moreover, these acylcarnitines have been found to increase reactive oxygen species (ROS) production, cyclooxygenase-2 expression, and stimulate mediators of the inflammatory pathway response, like JNK and ERK kinases [60]. In addition to elevated levels of acylcarnitines, the patients included in the study also demonstrated slightly elevated levels of IL-6 and TNF-α prior liver transplantation.

The metabolites PC.aa.C32.0, PC.aa.C34.1, PC.aa.C38.6, PC.aa.C40.5, and PC.aa.C40.6 are all phosphatidylcholines (PCs), constituting important components of cell membranes, and they play a decisive role in cellular signaling [63]. The perturbation of PC homeostasis is well known to cause cell death and increase the degree of liver pathology [64,65]. An impaired PC biosynthesis has already been observed in a number of pathological diseases of the liver in humans, among others, also in patients with liver failure [63,66,67]. This is consistent with our results; moreover, the concentrations of these metabolites in plasma normalized within the first hours after transplantation, indicating that the transplanted livers are functioning appropriately and normalizing the body’s homeostasis. This is also reflected by the significantly decreasing and normalizing cytokine levels in the transplanted patients (Table 2).

The significantly reduced levels of PCs (PC.aa.C34.1, PC.aa.C38.6, PC.aa.C40.5, PC.aa.C40.6) may be explained with two mechanisms: first, a decreased synthesis of PCs, and second, an increased catabolism [68]. Hepatic phosphatidylcholine is formed via the CDP–choline pathway (approx. 70% of total biosynthesis) and via the methylation of phosphatidylethanolamine (PE, approx. 30% of total biosynthesis) by the enzyme PE-N-methyltransferase (PEMT) [63]. Hypoxia and impaired oxidative phosphorylation may cause an insufficient formation of ATP and CDP–choline and thus lead to disturbance of the CDP pathway [68]. Likewise, decreased PEMT activity was also observed when the liver is damaged, resulting in a decreased synthesis of PCs [69]. Another possible cause for the change in this PC is the ELOVL metabolism, which comprises a series of elongases for long-chain fatty acids (LC-FAs) and is also important for the synthesis of unsaturated FAs. The synthesis of C18-FA, which is the first step of ELOVL metabolism, is mediated by the NADPH-dependent ELOVL6 [70]. As a result of oxidative stress, ELOVL6 activity is inhibited, leading to a decrease in PC with long-chain unsaturated FAs (i.e., PC.aa.C34.1, PC.aa.C38.6, PC.aa.C40.5 and PC.aa.C40.6), whereas PC with medium-chain saturated fatty acids (i.e., PC.aa.C32.0, composed of C14-, C16- or C18-FA) are increased [71,72]. Importantly, a similarly altered fatty acid composition was also found in cirrhotic patients prior to liver transplantation, supporting our hypothesis [73]. Nevertheless, the reduction in PC with long-chain unsaturated FAs may also be caused in part by lipid peroxidation, and we therefore hypothesize a combination of both mechanisms.

When PC homeostasis is perturbed, either by an impaired production, an increased catabolism of PCs or a combination of both, the activation of repair and compensation mechanisms is likely to occur first. If these repair and compensation mechanisms are inadequate or even fail, cell death seems to be the easiest way out [64,74]. So far, there is limited information on which specific pathway is involved in apoptosis induced by PC perturbation. As the SM metabolism is often partly involved in the process of apoptosis, an inhibition of PC synthesis may lead to a decrease in SM and an accumulation of pro-apoptotic ceramides [64]. Decreased concentrations of SM (SM.C16.1 and SM.C20.2) were also demonstrated in our study. SMs are hydrolyzed from membranes by specific enzymes, including sphingomyelinases, to generate ceramide [75].

In particular, TNF-α is known to activate sphingomyelinases, both in cell culture models and in vivo [76,77,78,79]. The patients included in the study demonstrated slightly elevated levels of TNF-α preoperatively, too.

The degradation of sphingomyelin into ceramide contributes to cell death induction [75,80]. In fact, it has been demonstrated that disturbed sphingolipid homeostasis leads to hepatocellular death and is therefore also associated with several liver diseases such as steatohepatitis, cirrhosis and drug toxicity [81,82,83]. Considering that inflammatory processes and oxidative stress, among others, are involved in the generation of DILI, altered sphingomyelin levels may be able to model hepatocellular injury, too.

As the metabolic biomarker set is suitable to reveal hepatocellular injury in vivo, it was verified in this study whether it is also capable of displaying cell damage in simple in vitro cell models. This would allow the use of the biomarker set as translational biomarkers and as a basis for in vitro models for metabolic analysis in drug toxicity testing.

APAP and other hepatotoxic drugs were already demonstrated to cause increased levels of acylcarnitines and sphingolipids in mouse models [84,85,86], cell culture models [16,87] and humans [88]. We used APAP exposure as a surrogate for hepatotoxicity in our cell models, and both PHH as well as HepG2/C3A hepatocytes demonstrated changes in the concentrations of some metabolites contained in the biomarker set found in vivo. However, not all metabolites found to be changed in vivo were also altered in the cell culture models, and differences between both cell culture models were observed.

When analyzing the PHH cells, 147 analyzed metabolites had to be excluded as they showed a clearly different behavior between the two PHH donors. This can be explained by a considerable variability between primary human hepatocyte preparations from different donors. Furthermore, cells from different donors may vary in the expression of drug-metabolizing enzymes and therefore may not respond uniformly to the identical treatments [33,34]. Indeed, such variability may be reduced by increasing donor numbers, although in our case, this was not possible due to limited availability, highlighting a clear disadvantage of primary hepatocytes. From the remaining metabolites, only seven showed the same behavior as observed in the patients. For the analysis of HepG2/C3A, 15 of the 35 metabolites demonstrated significant changes.

Both PHH and HepG2/C3A hepatocytes are used in pharmaceutical and toxicological studies. The fact that some drug-metabolizing enzymes are generally expressed at lower levels or are even undetectable in permanent hepatocytes is often mentioned as the major restriction and has been addressed in various studies [36,37,38]. In these studies, however, the gene expression of different cell cultures was investigated at baseline levels without exposure to xenobiotics. Interestingly, another study revealed that in response to the administration of drugs, changes in gene expression in permanent hepatocytes are more similar to those of primary human hepatocytes compared to non-exposure to compounds [89].

Furthermore, it has been already demonstrated that the subclone C3A exhibits biosynthetic capabilities similar to primary human hepatocytes [24]. It is therefore quite possible that HepG2/C3A cells have an appropriate ability to carry out biotransformation reactions and thus can be used as in vitro models for the investigation of endogenous metabolic reactions to drugs and other substances [90,91,92,93].

The hepatotoxic potential of acetaminophen in vivo is widely known, and APAP accounts for a significant proportion of acute liver failures due to overdosage [11]. For fluconazole, anidulafungin and caspofungin, rare cases of hepatotoxic events in vivo have been reported [94,95,96]. The hepatotoxic potential of rocuronium and sufentanil in vivo is not known yet [97].

Using the metabolic biomarkers of hepatocellular injury determined in vivo, a hepatotoxicity was confirmed in both PHH and HepG2/C3A hepatocytes as a result of APAP exposure in toxic concentrations. Comparable effects were also shown after the exposure with ROC, ANIDU and CASPO but not for all of the metabolites as after APAP exposure. However, for the exposure with SUF and FLUCO, only slight changes and a small number of significantly changed metabolites had been observed.

ROC demonstrated a concentration-dependent hepatotoxic potential in in vitro HepG2/3A cell models with decreased cell counts and impaired functionality in response to the exposure [98]. In the metabolic analysis in our study, ROC also demonstrated significant changes in some of the metabolites of the biomarker set, both acylcarnitines and phosphatidylcholines. However, a dose dependence could not be clearly observed here. The exact mechanism of ROC causing hepatotoxicity is still unclear but may be related to oxidative stress and mitochondrial impairment.

The hepatotoxic potential of ANIDU and FLUCO in in vitro HepG2/3A cell models had been demonstrated in viability tests in one study, whereas for CASPO, only mild hepatotoxic potential has been observed there [99]. In the study mentioned, mild levels of hepatotoxicity were demonstrated for clinical plasma concentrations (C_max_). As drug concentrations increased, however, ANIDU and FLUCO demonstrated dose-dependent liver injury. Nevertheless, in our study, no hepatocellular injury can be detected based on the metabolic biomarkers in response to the exposure with FLUCO even at higher concentrations. Other studies demonstrated a hepatotoxicity of FLUCO in HepG2 cells only with concentrations > 100 µM [100,101]. Considering that we used a lower concentration (81.63 µM), it can be assumed that there was no hepatotoxicity in our in vitro HepG2/C3A.

For ANIDU and CASPO, no further evidence for in vitro hepatotoxicity after exposure as a single compound could be found. In our study, some metabolites of the biomarker set had been altered in response to the exposure with ANIDU and CASPO with similar changes for C_max_ and 10× C_max_. Reactions of acylcarnitines were found to be comparable to the responses to APAP exposure, whereas phosphatidylcholine levels were altered contrarily.

The exposure with SUF led to a slight change and a small number of significantly changed metabolites. In contrast, a concentration-dependent hepatotoxic potential of SUF was demonstrated in in vitro HepG2/3A cell models [102]. Nevertheless, the mechanisms for the hepatoxicity of SUF are still unclear.

Considering that not all metabolites identified in vivo also demonstrated a response to hepatotoxic drug exposure in HepG2/C3A cells, only those metabolites that were altered similar in vivo and in vitro should be considered to serve as a translational metabolic biomarker set. A study using HepG2 cells to investigate a predictive model for discriminating between non-toxic and hepatotoxic drugs also revealed increases in acylcarnitines, as presented here. However, the increase in acylcarnitines was only significant for drugs associated with causing oxidative stress and phospholipidosis [103]. The biomarker set identified in our study includes metabolites belonging to the classes of acylcarnitines, phosphatidylcholines and a sphingomyelines. This could be advantageous as it allows to display different cellular reactions and therefore may improve the predictability of toxic drug reactions in cell culture models. For drug toxicity screenings based on these metabolic biomarkers, the combination of several cell culture models (PHH, permanent cell lines (such as HepG2, HepaRG, L-02, Huh-7), 3D spheroid cultures) or co-cultures of different cell types should be considered, and it should be investigated whether an even better prediction of hepatocellular injury can be achieved in this way.

The metabolic biomarker set identified in this study seems promising for the in vivo diagnostic of hepatocellular injuries of different pathogenesis. Containing metabolites from different classes, the set allows a wide range of cellular metabolic pathways to be covered. The potential of the set is also shown by the fact that the plasma concentrations of the metabolites were tending to return to concentrations comparable to healthy controls within the first few hours after liver transplantation. This implies that the restored hepatic function induces the normalization of the bodies’ homeostasis.

## 4. Materials and Methods

### 4.1. Patients, Collected Samples, Measurement of Cytokines

Serum and plasma samples of 11 patients suffering from acute-on-chronic liver failure with different pathogenesis (liver-transplantation group), serum samples of 21 post-OP control patients (patients with surgical intervention for different diagnosis/without liver insufficiency; postoperative-control group) and serum samples of 14 healthy control patients were collected at the university clinic Rostock. Recruitment of the patients occurred from March 2006 to December 2009. The survival, laboratory parameters, and clinical characteristics of patients were obtained and are displayed in Table 1. Samples were collected from liver insufficiency patients before and at several time points (2 h, 12 h, 24 h) after liver transplantation (LT). The sampling of post-OP control patients took place either at 2 h, 72 h or 168 h after the surgical intervention. All patients received sufentanil and rocuronium intraoperatively as repeated single doses.

All samples are displayed in Figure 5. Informed consent was obtained from all patients. The study was approved by the ethics committee of Rostock University (reg-nr. II HV 01/2006). Blood samples were collected from the central venous or arterial line with heparin as anticoagulant or as serum probes. After taking the anticoagulated blood, the samples were then centrifugated for 15 min at 4.000 rpm. The collected plasma was aliquoted into 1.5 mL reaction vessels and stored at −80 °C until further use. All plasma samples were processed at 2–8 °C within 90 min after collection.

In the liver-transplantation group, the cytokines TNF-alpha, IL-6 and IL-10 IL-1 beta were measured in patients’ serum with commercial ELISA kits as described by the supplier (BioSource International, Camarillo, CA, USA).

### 4.2. Culture and Treatment of Cells

PHHs were obtained from two different donors after partial liver resection (PRIMACYT Cell Culture Technology GmbH, Schwerin, Germany) and cultured in the HHMM cell culture medium. For the donors’ characteristics, see Appendix A. HepG2/C3A cells (ATCC HB-8065) were cultured in Dulbecco’s modified Eagle’s medium (DMEM, GIBCO Life Technologies, Darmstadt, Germany) supplemented with 10% fetal bovine serum (FBS, PAA Laboratories, Pasching, Germany), 1% of 200 mM L-glutamine (PAA), and 1% of antibiotics solution (penicillin G: 10.000 IE/mL/streptomycin: 10 mg/mL; PAA). All cell lines were cultivated at 37 °C in a humidified atmosphere of 5% CO_2_.

Primary and permanent cell cultures were treated with different concentrations of six approved drugs; plain culture medium was used as a vehicle control (Table 4). The respective mean plasma levels of the different drugs after the injection of an intravenous therapy (C_max_) were used as the lowest concentrations [104,105,106,107]. In addition to C_max_, also higher concentrations (10× C_max_) of the drugs were investigated. Cells were seeded in 24-well plates at 5 × 10^5^ (HepG2/C3A) or 3 × 10^5^ (PPH) cells/well and treated for 72 h, following the addition of fresh media, and treatment continued for another 72 h.

### 4.3. LC/MS-Based Metabolomics

Metabolite extraction and measurement was performed using a NextreraXR HPLC (Shimadzu, Kyoto, Japan) connected to a 4000 QTRAP mass spectrometer (ABI Sciex, Framingham, MA, USA) as previously described [108]. In brief, cell pellets were washed three times with ice-cold PBS, resuspended at 2.5 × 10^7^ cells/mL in ice-cold ethanol containing 15% (*v*/*v*) 10 mM K_2_PO_4_, pH 7.5 and lysed by three sonication–freeze cycles. Human plasma samples were thawed on ice and centrifuged at 10.000 rcf for 10 min at 4 °C prior to measurement. Metabolite quantification was performed using the AbsoluteIDQ p180 Kit (Biocrates Life Science AG, Innsbruck, Austria) according to the manufacturer’s instructions. For a detailed list of metabolites analyzed, see Appendix A.

### 4.4. Statistical Analysis

The results of measurements of cytokines and clinical parameters are expressed as the median with the 0.25–0.75 quartile. Nonparametric analyses were used after (negative) testing of normal distribution (with the Kolmorgorov–Smirnov test; SPSS 29.0, SPSS Inc., Chicago, IL, USA). Statistical significance was analyzed with the Kruskal–Wallis test, the Mann–Whitney U-test, the Friedman test, and the Wilcoxon test. Statistical significance was considered when the *p*-value was <0.05.

Statistical analysis of the metabolic biomarker results was performed with the statistic software R (version 4.0) [109] and R package “car” (version 3.0-10) [110] to identify changes in metabolite concentrations between patients/drug treatment groups and control groups. Log_2_ transformation was performed on the obtained concentration data for normalization. Plate effects were removed by nonparametric batch correction and verified by three-way ANOVA analysis with interaction for PHH experiments (plate, drug treatment, cell donors), respectively, two-way ANOVA for patients and HepG2/C3A experiments (plate, patient groups/drug treatment). *p*-values were false discovery rate adjusted (as per Benjamin–Hochberg procedure). Data were further investigated with two-way ANOVA (treatment, cell donors) for PHH cell culture experiments, and only metabolites showing no significance between the donors were investigated further. The patient and HepG2/C3A experimental data were investigated with one-way ANOVA (treatment). A pairwise *t*-test was performed, and pairwise log fold changes of metabolite concentrations between patient/treatments and control groups were determined. Metabolites with absolute log fold changes of 1 (corresponding to changes of 100%) and adjusted *p*-values below 0.05 were considered relevant.

## 5. Conclusions

The aim of the presented study was to identify translational biomarkers for acute liver injury in human patients that can serve as biomarkers for hepatocellular injury in vivo and in vitro in simple cell models for drug toxicity testing.

We identified a promising metabolic biomarker set containing 15 metabolites (C2, C3, C3.OH, C4, C3.DC, C4.OH, C4.1, C5, C5.DC, C6.OH, C8, C10, PC.aa.C34.1, PC.aa.C38.6, PC.aa.C40.5, PC.aa.C40.6 and SM.C16.1) demonstrating significant changes for hepatocellular injuries in vivo and in vitro. The metabolic biomarker set includes several acylcarnitines, phosphatidylcholines and a sphingomyeline that are, in altered levels, associated with major disturbance of cell biochemistry and the impairment of key cell metabolic functions associated with the energy production, elimination of waste products, antioxidant defense mechanisms, and maintenance of cell structure and homeostasis.

The results obtained suggest that simple in vitro cell models have the capability for displaying metabolic responses in biomarkers for hepatocellular injury in the course of the treatment with (hepatotoxic) drugs and therefore can serve as a basis for in vitro models for metabolic analysis in drug toxicity testing.

The metabolic biomarker set should furthermore be evaluated for its potential to serve as diagnostic screening test in vivo for patients with the risk of developing hepatocellular injury, respectively, liver failure as well as for monitoring treatment responses to therapies.

## Figures and Tables

**Figure 1 ijms-25-13739-f001:**
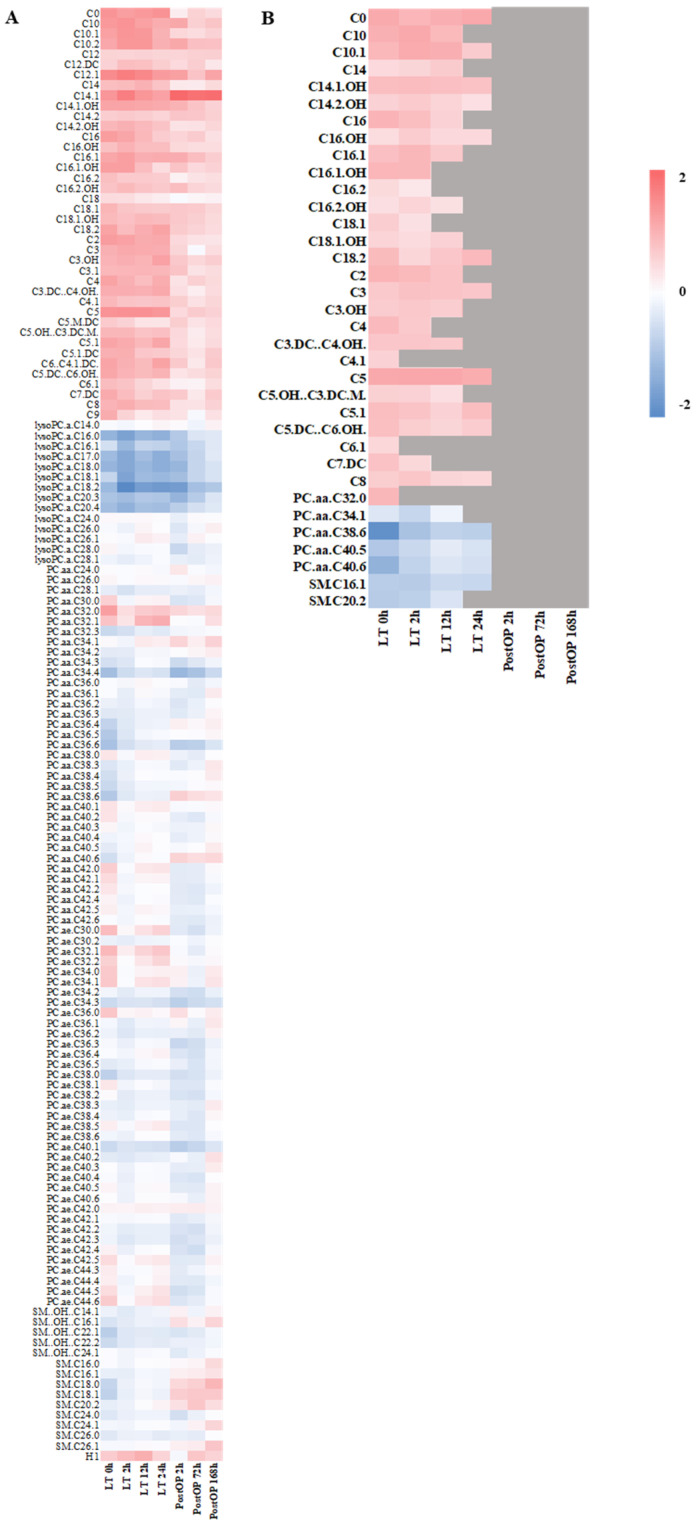
Metabolites changed in patients with an acute-on-chronic liver failure before and after liver transplantation and identified metabolic biomarker set for hepatocellular injury. (**A**) Mean log2 fold changes of metabolite concentrations in patients with an acute-on-chronic liver failure before (LT 0 h) and at several time points after liver transplantation (LT 2 h, LT 12 h, LT 24 h) as opposed to post OP control patients (post-OP 2 h, post-OP 72 h, post-OP 168 h) each in comparison to healthy controls (for a detailed heatmap of log2 fold changes for each patient, see Appendix A). (**B**) Detailed view of mean values of metabolites found with significant (*p* < 0.05) log2 fold changed concentrations in patients with hepatocellular injury as opposed to post-OP patients, each in comparison to healthy control patients potentially serving as a translational metabolic biomarker set for hepatocellular injury. Metabolites that were not significantly changed are highlighted in gray.

**Figure 2 ijms-25-13739-f002:**
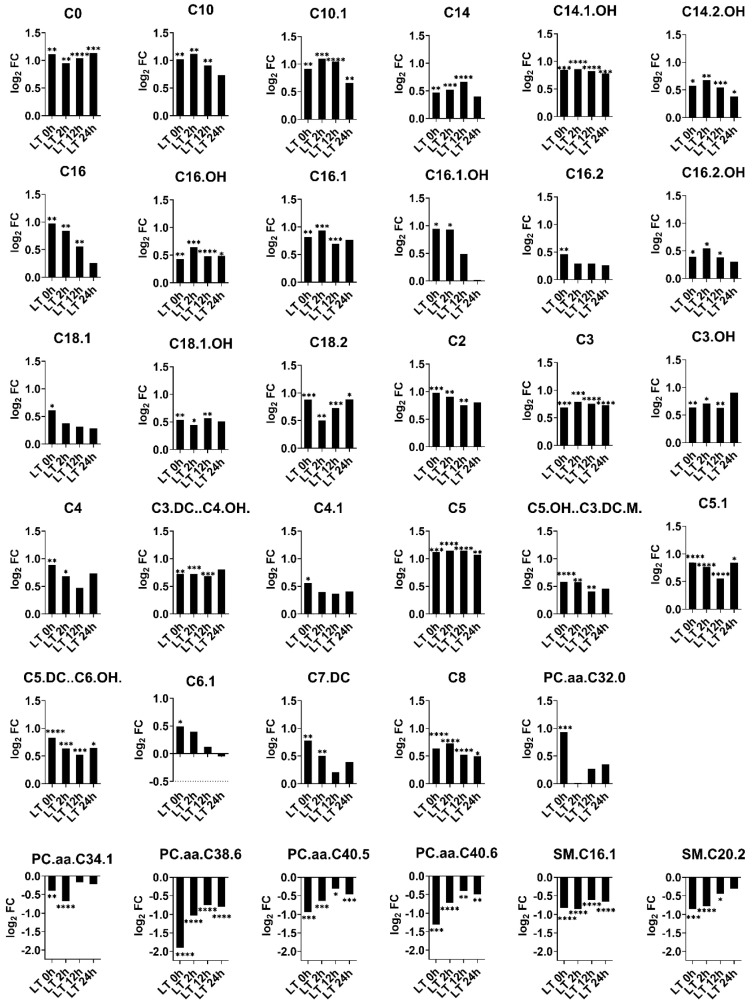
Log2 fold changes of metabolites significantly changed in patients with an acute-on-chronic liver failure before liver transplantation (LT 0 h) as opposed to log2 fold changes of metabolites after liver transplantations (LT 2 h, LT 12 h, LT 24 h). Significant changes are marked (* *p* < 0.05, ** *p* < 0.01, *** *p* < 0.001, **** *p* < 0.0001).

**Figure 3 ijms-25-13739-f003:**
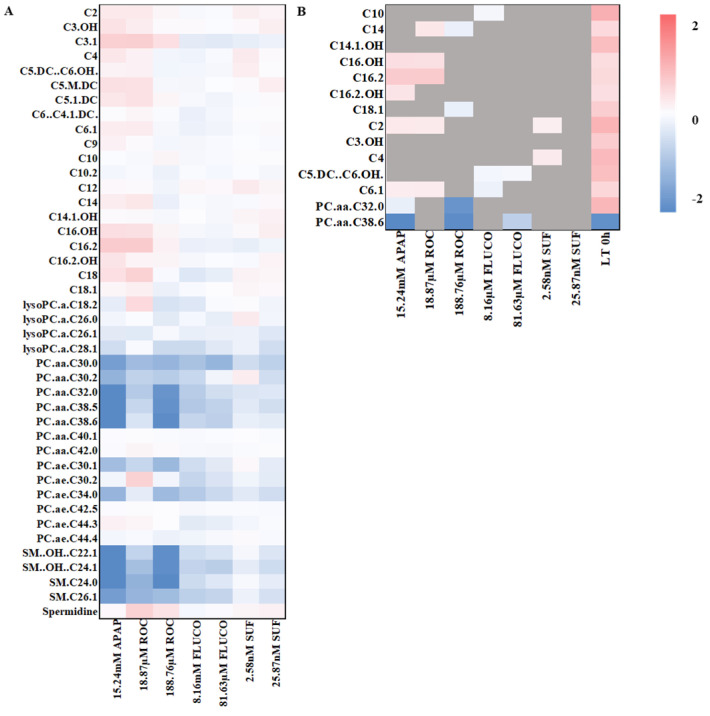
Metabolites changed in primary human hepatocytes in response to the exposure with several drugs (**A**) Mean log2 fold changes of metabolite concentrations in primary human hepatocytes from two donors in comparison to vehicle control (for a detailed heatmap of log2 fold changes for each sample, see Appendix A). (**B**) Detailed view of log2 fold changes of metabolites found with significant (*p* < 0.05) log2 fold changed concentrations in PHH from two donors in comparison to vehicle control opposed to log2 fold changes in patients with an acute-on-chronic liver failure before liver transplantation. Metabolites that were not significantly changed are highlighted in gray.

**Figure 4 ijms-25-13739-f004:**
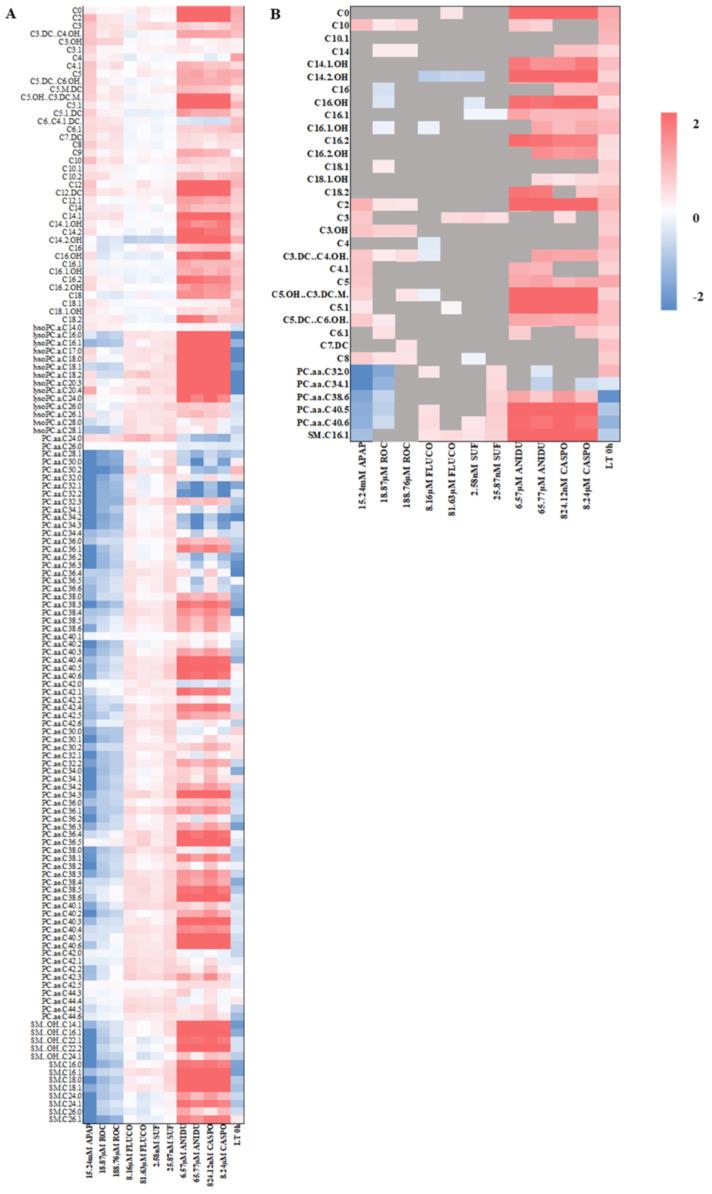
Metabolites changed in HepG2/C3A hepatocytes in response to the exposure with several drugs. (**A**) Mean log2 fold changes of metabolite concentrations HepG2/C3A in comparison to vehicle control opposed to log2 fold changes in patients with an acute-on-chronic liver failure before liver transplantation (for a detailed heatmap of log2 fold changes for each sample, see Appendix A). (**B**) Detailed view of log2 fold changes of metabolites found with significant (*p* < 0.05) log2 fold changed concentrations in HepG2/C3A in comparison to vehicle control as opposed to log2 fold changes in patients with an acute-on-chronic liver failure before liver transplantation. Metabolites that were not significantly changed are highlighted in gray.

**Figure 5 ijms-25-13739-f005:**
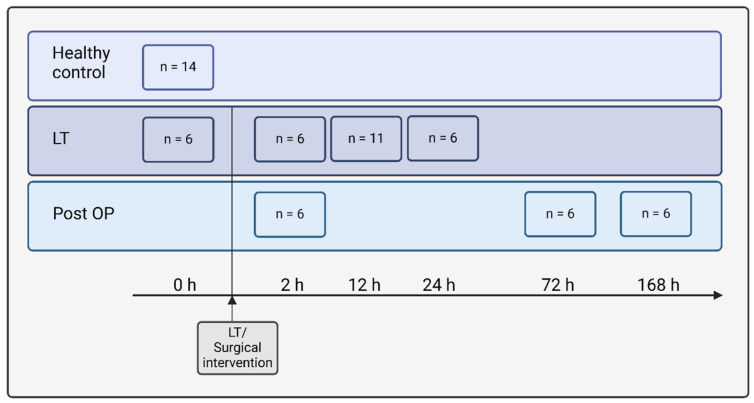
Flowchart of patient cohort. The patient cohort includes samples of patients with liver insufficiency before and at several time points (2 h, 12 h, 24 h) after liver transplantation (LT), samples of patients with different surgical interventions at different time points (2 h, 72 h, 168 h) after the intervention (post OP) and samples of healthy controls.

**Table 1 ijms-25-13739-t001:** Baseline characteristics of patients and results of laboratory parameters and results immediately after surgical treatment and admission to ICU in the liver-transplantation group (LTG, *n* = 11) and the postoperative-control group (PCG, *n* = 21); (median/0.25–0.75 quartile).

Parameter/Values	Liver-Transplantation Group (LTG, *n* = 11)	Postoperative-Control Group (PCG, *n* = 21)
**Age**	**53** **(46/59)**	**67** **(56/70)**
**Female/Male**	**(10/1)**	**(17/4)**
**Lactate** (mmol/L)	**2.5 *** (1.8–4.0)	**1.1** (0.8–1.2)
**Bilirubin** (µmol/L)	**86.9 *** (66.9–161.0)	**14.1** (11.3–18.9)
**Ammonia** (mmol/L)	**67.3 *** (51.9–85.3)	**33.7** (28.1–39.3)
**Creatinine** (µmol/L)	**117.0** (104.9–144.0)	**80.4** (72.2–90.9)
**Urea** (mmol/L)	**9.2** (6.3–18.0)	**4.6** (3.4–5.8)
**PCT** (ng/mL)	**1.08** (0.31–1.64)	**0.61** (0.29–0.98)
**Leukocytes** (GpT/L)	**6.3** (4.8–12.7)	**9.0** (7.6–70.7)
**Thrombocytes** (GpT/L)	**95 ^#^** (77–99)	**171** (132–229)
**Prothrombin time** (%)	**44 *** (41–48)	**91** (86–95)
**APACHE II**	**29 *** (28–32)	**9** (8–12)
**SOFA**	**16 *** (15–17)	**2** (1–4)
**CHILD PUGH Score**	**12** (11–13)	**-**

^#^ *p* < 0.01 vs. control group (CG); * *p* < 0.001 vs. control group (CG); APACHE: Acute Physiology and Chronic Health Evaluation; PCT: procalcitonin; SOFA: sepsis-related organ failure assessment.

**Table 2 ijms-25-13739-t002:** Cytokines values at inclusion and after surgical treatment and admission to ICU in the liver-transplantation group (LTG, *n* = 11); (median/0.25–0.75 quartile).

Cytokine (pg/mL)	At Inclusion (Before Liver Transplantation)	After Surgical Treatment and Admission to ICU			
		2 h	12 h	24 h	168 h
**IL-6**	**16.4** (9.8–24.7)	**185.5** (108.4–944.3) *	**62.5** (33.7–123.3) +	**41.6** (25.3–59.1) +	**58.0** (25.7–77.5) +
**IL-10**	**2.5** (2.5–7.8)	**938.5** (356.5–1449.3) *	**27.0** (13.1–86.8) * +	**12.4** (10.2–16.4) +	**48.7** (17.0–77.9) * +
**TNF-alpha**	**15.3** (9.0–16.8)	**18.6** (16.1–24.4) *	**16.5** (10.7–20.8) +	**15.4** (12.7–23.7)	**14.9** (13.3–28.1) +

IL: interleukin; TNF: tumor necrosis factor; statistically significant (*p* < 0.05): * vs. before liver transplantation; + vs. 2 h after liver transplantation.

**Table 3 ijms-25-13739-t003:** Overview of metabolites with significant log2 fold changes in patients with an acute-on chronic liver failure before transplantation and in HEPG2/C3A cell culture model in response to the exposure with several drugs at C_max_ and 10× C_max_. ↑—increase, ↓—decrease.

Metabolite	LT	HepG2/C3A										
		APAP	ROC		FLUC		SUF		ANIDU		CASPO	
			C_max_	10× C_max_	C_max_	10× C_max_	C_max_	10× C_max_	C_max_	10× C_max_	C_max_	10× C_max_
C0	↑					↑			↑	↑	↑	↑
C10	↑	↑	↑	↑					↑	↑		
C10.1	↑											
C14	↑		↑	↑							↑	↑
C14.1.OH	↑								↑	↑	↑	↑
C14.2.OH	↑				↓	↓	↓		↑	↑	↑	↑
C16	↑		↑								↑	↑
C16.OH	↑		↑				↑		↑	↑	↑	↑
C16.1	↑						↑	↑	↑	↑	↑	↑
C16.1.OH	↑		↑		↑					↑	↑	↑
C16.2	↑								↑	↑	↑	↑
C16.2.OH	↑									↑	↑	↑
C18.1	↑		↑									
C18.1.OH	↑									↑	↑	↑
C18.2	↑								↑	↑		↑
C2	↑	↑	↑	↑					↑	↑	↑	↑
C3	↑	↑				↑	↑	↑			↑	
C3.OH	↑	↑	↑	↑								
C4	↑				↑							
C3.DC…C4.OH.	↑	↑	↑	↑	↑					↑	↑	↑
C4.1	↑	↑							↑	↑		
C5	↑	↑							↑	↑	↑	↑
C5.OH…C3.DC.M.	↑			↑	↑				↑	↑	↑	↑
C5.1	↑								↑	↑	↑	↑
C5.DC…C6.OH.	↑	↑	↑						↑	↑	↑	↑
C6.1	↑		↑						↑			↑
C7.DC	↑			↑								
C8	↑	↑	↑	↑			↑					
PC.aa.C32.0	↑	↓	↓		↓			↓		↓		
PC.aa.C34.1	↓	↓	↓					↓		↓		↓
PC.aa.C38.6	↓	↓	↓					↓	↑	↑	↑	↑
PC.aa.C40.5	↓	↓	↓		↓			↓	↑	↑	↑	↑
PC.aa.C40.6	↓	↓	↓		↓		↓	↓	↑	↑	↑	↑
SM.C16.1	↓	↓			↓	↓	↓	↓	↑	↑	↑	↑
SM.C20.2	↓											

**Table 4 ijms-25-13739-t004:** Compounds and concentrations selected for in vitro cell culture models. The compounds include rocuronium, sufentanil, fluconazole, anidulafungin and caspofungin at Cmax and 10× Cmax. Acetaminophen was used at 10× Cmax as positive control to induce DILI.

Class	Compound	Abbreviation	C_max_	10× C_max_
Negative control	Culture medium			
Positive control	Acetaminophen	APAP		15.24 mM
Test compounds	Rocuronium	ROC	18.87 µM	188.76 µM
	Sufentanil	SUF	2.58 nM	25.87 nM
	Fluconazole	FLUCO	8.16 µM	81.62 µM
	Anidulafungin	ANIDU	6.57 µM	65.77 µM
	Caspofungin	CASPO	824.12 nM	8.24 µM

## Data Availability

The raw data supporting the conclusions of this article will be made available by the authors on request.

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
