# Peer review of "Metabolic Biomarkers of Liver Failure in Cell Models and Patient Sera: Toward Liver Damage Evaluation In Vitro"

_ijms, 2024, doi:10.3390/ijms252413739_

Round 1
Reviewer 1 Report
Comments and Suggestions for Authors
Thank you, Mr. Editor, for giving me the opportunity to review this manuscript. The authors have excelled in selecting the research topic, presenting the results clearly, and writing the study in a distinguished scientific manner
But only authors need update the references. Also, can clarify the rationale for selecting the number of patients as a sample that aligns with the research concept.?

Writing of the research needs some clarification to make it easier for the reader, especially in the results and discussion sections.
Reviewer 2 Report
Comments and Suggestions for Authors
In this study, changes in metabolite concentrations in plasma sample from patients suffering from liver failure with different pathogenesis have been evaluated and compared with those in primary and permanent human hepatocyte cultures in response to the treatment with several drugs. A time dependence of the changes after liver transplantation is reported. Only those metabolites that were altered similarly in-vivo and in-vitro are proposed to serve as a translational metabolic biomarker set, of about 15 molecules.
1. The MS was probably initially written for another journal, were the experimental section comes before Results. In its present form it sounds odd to have as the first sentence of the Results the following: “Only one patient of the liver-transplantation group died (in-hospital mortality) and all patients of the postoperative-control group survived”. The authors should at list first indicate the number of subjects and groups involved and introduce Table 1 for the description of the analyzed cohort.
2. Lines 176-178: the concept of “normalize” should be better explained. Indeed, at 168 h only the value of TNF-a goes back to pre-transplantation values (and it never increases too much). Instead, for IL-6 and Il-10 a decreasing trend is observed when moving from 2 to 12 and then to 24 h after transplantation; the values seem to increase again at 168 h.
3. Figure 1 provides info on a long list of molecules. It would be useful to identify meaningful log2(FC) (p-value <0.05) with an asterisk directly in this figure.
4. In section 4.1 a detailed of the pre-analytical procedures should be reported (see ISO 23118:2021 or doi: 10.1016/j.nbt.2019.04.004 for key parameters).
5. In general, the introduction and discussion would benefit from being shortened. This would make them more concise and to the point.
Minor points:
Line 145: biomarkers à biomarkers
Throughout the text: Use a consistent way of writing log2(FC) (log2, log-2 …)
Reviewer 3 Report
Comments and Suggestions for Authors
The authors attempt to define differential metabolomics in patients with ACLF versus healthy livers.
Overall we have seen an explosion of this type of research, which distinguishes differences in peripheral blood between cohorts. However, this lacks any clinical significance and has no validation cohort, meaning these results could be secondary to any number of findings. Unfortuantley I just do not feel this approach has enough scientific rigor at this time for publication.
Reviewer 4 Report
Comments and Suggestions for Authors
Manuscript: “Metabolic biomarkers of liver failure in cell models and patient sera: towards liver damage evaluation in vitro”
The manuscript submitted by Rentschler et al deals with a very important health issue that needs considerable attention. The authors provide an interesting study that provides some valuable information but still has some issues that need to be addressed before their results could be translated into effective evaluation of liver damage in viro.
Major comments:
i) The authors made an analysis of the toxicity of drugs in PHH cells from two distinct donors. The simple use of two donors caused the elimination of 147 metabolites from the initial list of 188. If this was the case with two donors, which would be the scenario of using a higher number of donors? An eventual elimination of the whole list due to significant different behaviour between each of the PHH cells derived from each donor? This donor dependency/variability would ultimately render the analysis using PHH cells obsolete.
ii) The sentence “It is therefore quite possible that HepG2/C3A cells have an appropriate ability to carry out biotransformation reactions and thus can be used as in-vitro models for the investigation of endogenous metabolic reactions to drugs and other substances.” needs confirmation. We can understand that such could be the case but without effective confirmation and characterization of those biotransformation reactions there is not much support for their use.
Comments:
i) In the sentence “In response to hepatotoxic 238 APAP exposure, 6 metabolites of the biomarker set showed in the PHH cells a similar behavior as observed in the patients (C2, C6.1, C16.OH, C16.2, C16.2.OH, PC.aa.C38.6). 240 PC.aa.C32.0, however, was significantly altered in the opposite direction.”. How can the authors refer to a similar behaviour and immediately after say that the effects were in the opposite direction? If the effects were in the opposite direction the behaviour was opposite, not similar as the authors refer.
ii) In the discussion section, page 13 lines 391-406: I can understand the discussion around the decrease in levels of many PC’s (lines 391-400) but on the contrary I could not follow the authors explanation for the increase in PC.aa.C32:0.
iii) The authors refer that the set of potential metabolic biomarkers identified in the study is promising. As mentioned above, how meaningful is the identified set if only PMM cells from two donors were used. An increment of the number of PMM cell donors needs to be made before any conclusions can be drawn from this study.
Minor comments:
i) Page 2 line 63: “Besides the concerns”; the term Besides in the beginning of the sentence could cause the impression that the animal welfare is a less important issue than cost and time consumption. I suggest a rephrasing of the sentence to eliminate such impression.
ii) Page 3 line 124: segment of sentence “respectively lower levels of enzymes than found in vivo” makes no sense in the context of the written sentence.
iii) Page 3 line 138: instead of comma after ref. [23] needs period.
iv) Page 4 line 161: “and predominant female patients”; the authors must mean predominantly.
v) Page 9 line 263: Do the authors mean Figure 4A?
vi) Page 12 lines 313-314: there is repetition of “rare cases”.
vii) Page 12 lines 315-316: why the need for paragraph change? Why first does not appear immediately after “:”; it’s an enumeration.
viii) Page 13 lines 364-366: The sentence needs text reorganization.
ix) Page 13 line 371: The C12-ACs is considered long-chain? I have the impression that long chain would be C14 or higher.
x) Page 15 line 467: “there” should be substituted by they are.
xi) Page 15 line 480: “The absence respectively very low concentration”; what does it mean?
xii) Page 16 line 549: correct the word “phospholopidosis” to phospholipidosis.
Round 2
Reviewer 3 Report
Comments and Suggestions for Authors
My overall issue with methodology remains but if the editors and other reviewers feel this is worth publication then I am happy to step back.
Reviewer 4 Report
Comments and Suggestions for Authors
The authors replied satisfactorily to my comments.